# Interaction of Squid (*Dosidicus giga*) Mantle Protein with a Mixtures of Potato and Corn Starch in an Extruded Snack, as Characterized by FTIR and DSC

**DOI:** 10.3390/molecules26072103

**Published:** 2021-04-06

**Authors:** José Luis Valenzuela-Lagarda, Ramón Pacheco-Aguilar, Roberto Gutiérrez-Dorado, Jaime Lizardi Mendoza, Jose Ángel López-Valenzuela, Miguel Ángel Mazorra-Manzano, María Dolores Muy-Rangel

**Affiliations:** 1Centro Regional de Educación Superior de la Costa Chica, Universidad Autónoma de Guerrero, Cruz Grande 41800, Mexico; joseluislagarda@uagro.mx; 2Centro de Investigación en Alimentación y Desarrollo, A.C. Unidad Hermosillo, Hermosillo 83304, Mexico; rpacheco@ciad.mx (R.P.-A.); jalim@ciad.mx (J.L.M.); mazorra@ciad.mx (M.Á.M.-M.); 3Facultad de Ciencias Químico-Biológicas, Universidad Autónoma de Sinaloa, Culiacán 80013, Mexico; robe399@hotmail.com (R.G.-D.); jalopezvla@uas.edu.mx (J.Á.L.-V.); 4Centro de Investigación en Alimentación y Desarrollo, A.C. Unidad Culiacán, Culiacán 80110, Mexico

**Keywords:** squid, extruded, protein–starch interaction, melting temperature

## Abstract

The majority of snacks expanded by extrusion (SEE) are made with vegetable sources, to improve their nutritional content; it has been proposed to incorporate squid (*Dosidicus gigas*), due to its high protein content, low price and high availability. However, the interaction of proteins of animal origin with starch during extrusion causes negative effects on the sensory properties of SEE, so it is necessary to know the type of protein–carbohydrate interactions and their effect on these properties. The objective of this research was to study the interaction of proteins and carbohydrates of SEE elaborated with squid mantle, potato and corn. The nutritional composition and protein digestibility were evaluated, Fourier transform infrared (FTIR) and Differential Scanning Calorimetry (DSC) were used to study the formation of protein–starch complexes and the possible regions responsible for their interactions. The SEE had a high protein content (40–85%) and biological value (>93%). The melting temperature (Tm) was found between 145 and 225 °C; the Tm values in extruded samples are directly proportional to the squid content. The extrusion process reduced the amine groups I and II responsible for the protein–protein interaction and increased the *O*-glucosidic bonds, so these bonds could be responsible for the protein–carbohydrate interactions.

## 1. Introduction

Snacks expanded by extrusion (SEE) are foods with great acceptance by the consumer; these snacks are manufactured based primarily on corn, wheat, potatoes and rice. In general, SEE are high in carbohydrates, and some are fried in oils and have low protein content [1]. To improve the nutritional properties of this type of food, it has been proposed to incorporate products of marine origin as a source of protein of high biological value; one of the considered products is the giant squid (*Dosidicus gigas*) that is endemic to the Eastern Pacific and is abundant in Mexico [2]. In 2016, the giant squid reached the sixth national position (Mexico) in the volume of fishery products [3]; more than 90% of it was destined for fresh consumption as a source of low-cost protein. Recently, the incorporation of the muscle of the fishing species in extruded products has been evaluated; incorporation of starch is necessary to achieve acceptable physical properties and commercial competitiveness of SEE with animal protein [4]. During the production of SEE, multiple interactions occur between various ingredients (carbohydrates, proteins and water) that make up the extruded matrix and determine the quality of the product [5]. The proteins contained in marine foods are mainly of the fibrillar type with “compact” conformation and thus have a limited capacity to form three-dimensional structures that allow retention of the water vapor generated during the extrusion process [6].

Considering these properties, products made from proteins of animal origin have low values of expansion index (EI) and high values of bulk density (BD); these variables indicate quality standards of the expanded products and are determinants for acceptance by the consumer. To improve these characteristics, studies have been carried out to determine the effect of a combination of proteins of marine origin with various plants. A study by Shahmohammadi et al. [7] evaluated the effect of the content of silver carp (*H. molitrix*) combined with corn flour on the expansion index and the protein content in the extruded product and found that an increase in the fish content significantly reduced the EI. Several studies have shown that it is possible to prepare snacks based on mixtures of starch with crab [8], golden fish [9], red tilapia [10], shrimp [11] and fish-squid [12]; however, generally, these products have a poor expansion capacity (<1.5) due to the high proportions (>30%) of animal protein [8,11]. However, a recent study showed that it is feasible to obtain expanded snacks by extrusion with good expansion, density and texture characteristics from a mixture of squid/potato/corn flours including up to 60% of the squid meal [4]. Strong interactions between the main macromolecules (protein and starch) present in this extruded matrix can explain this phenomenon and the study of these interactions can significantly impact the development of expanded snacks by extrusion incorporating high levels of various proteins of animal origin in their composition.

Various methodologies have been proposed to study the protein–starch interactions that occur during the extrusion process including differential scanning calorimetry (DSC) and Fourier transform infrared (FTIR) spectroscopy. Two studies [13,14] evaluated the effect of incorporating milk protein concentrates into a corn starch matrix on the microstructural characteristics of a third-generation expanded snack; the authors reported that the enthalpy values of the extruded samples were lower (0.33–2 J/g) than those of the raw materials (8.0–12.13 J/g). This behavior was attributed to the fact that the extrusion process produces degraded materials that require less energy for their conformational changes. Similarly, Teoh et al. [15] conducted a study on the influence of matrix type (corn flour, corn gluten and native starch) on the calorimetric properties of expanded snacks by extrusion and concluded that the extrusion process promotes a reduction in the enthalpy values in the samples. This behavior is in agreement with reports of various authors on evaluation of the effect of the extrusion process on native starches [16,17].

Widjanarko et al. [5] conducted an FTIR study of mixtures of isolates of soy protein (ISP), wheat (IWP), maize (ICP) and glucomannan from konjac (Amorphophallus konjac) (KGM) for production of a food bar and reported infrared spectra that showed characteristic bands for each of the food components: 3460 cm^−1^ for ISP, 3432 cm^−1^ for IWP and 3400 cm^−1^ for ICP; these bands corresponded to vibrations of the -OH type bonds [5]. These researchers also observed that the spectra of mannose and glucose residues disappeared, which means that these residues of the KGM granules can interact with the protein isolates.

Thus, the objective of this work was to study the interaction of the squid mantle protein (*Dosidicus gigas*) with the starch of the potato and corn flours in the extrusion-expanded snack obtained from these ingredients using the DSC and FTIR techniques

## 2. Results

### 2.1. Proximal Analysis

The incorporation of giant squid allowed the attainment of SEE, with protein content of up to 62.8% when mixtures with 60% of squid were used in contrast with the protein content of snacks made with potatoes (17.2%) and corn (16.5%) (Table 1). Significant increase (*p* < 0.05) in the protein content is the result of the addition of squid, which is mostly composed of protein [4]. Snacks prepared with potatoes and corn had the highest carbohydrate content (65.8 and 66.2%, respectively) without significant differences between them *(p* > 0.05) due to the similar composition between both flours [18]. Similar values were reported by Onwulata et al. [19] in expanded snacks by extrusion made from potato and corn flours.

### 2.2. In Vitro Protein Digestibility (IVPD)

The F1 (inclusion level of squid (F1 = 01, 02, 40, 60, 80, 100% squid)) and F2 factors (sample without extruding and extruded) have a significant effect (*p* < 0.05) on IVPD and calorimetric properties. However, their interaction does not have a significant effect on these variables which is why only the main effects of F1 and F2 were analyzed in the discussion of the results. The highest values of IVPD (97.68) were obtained in the mixtures with 100% squid (Figure 1), which can be attributed to greater digestibility of the squid proteins compared to the protein–carbohydrate matrix. The data of Figure 1 indicate that the extrusion process significantly increased (*p* > 0.05) IVPD of the samples with 100% of potato and 100% of corn; however, addition of squid to the mixture did not result in a significant increase. This behavior can be attributed to the fact that plant samples have various anti-nutritional components that are eliminated during the extrusion process and their IVPD is enhanced [20,21]. In the case of samples containing squid, their fibrillar proteins do not interact with any anti-nutritional factors; hence, their IVPD values are excellent before and after the extrusion process.

Rathod and Annapure [22] found that the process of lentil extrusion significantly increased the IVPD from 39.4 to 88.7% which can be attributed to the fact that the combination of high temperatures, pressure and mechanical stress produced in the food due to extrusion reduces some anti-nutritional factors such as phytic acid, tannins, trypsin inhibitors and total polyphenol levels related to IVPD. Some natural chemical compounds known as “inhibitors” interact with the protein to form higher crossover complexes that affect their solubility forming protein complexes that are less susceptible to proteolytic degradation.

### 2.3. Calorimetric Properties

The data of Figure 2 indicate that calorimetric analyses of the extruded samples did not reveal the “glass transition” behavior which can be attributed to the extrusion effect that causes partial or total loss of the crystallinity due to the changes in structural factors in the starch and the protein–starch interaction that produces high molecular weight complexes. However, a melting point temperature transition between 145 and 225 °C was clearly visible in the analyzed samples (Figure 2). These data coincide with the study carried out by Heertjet et al. [23] in which they reported the formation of stable complexes of high molecular weight, which they attributed to the aggregation of carbohydrates to the protein and the inhibition of protein–protein interactions [23]. These changes are apparently the result of ionic forces, suggesting that the protein–carbohydrate interaction is due to electrostatic forces derived from the increase in net positive charge.

The Tm values of the extruded samples increased with increasing squid content. (Figure 2); this can be attributed to the fact that mollusk protein structures are mostly fibrillar with β-sheet configuration and their melting temperature is higher than that of the structure of the starch present in potatoes and corn [23]. The Tm values of the samples without extruding were in the range from 140 to 227 °C for squid contents of 40 and 100%, respectively. These values coincide with the numbers reported by Latza et al. [24] in a study on thermoplastic properties of high protein content materials obtained from giant squid suction teeth that had melting temperatures from 150 to 220 °C.

Melting temperatures of the samples with 40, 60 and 80% of squid increased significantly with the extrusion process but not for the samples with 100% squid (Figure 2). This can be attributed to the fact that in the samples that contain squid–potato–corn, interactions between the squid proteins and the starch present in the vegetable flours are favored, resulting in the generation of compounds with more complex structures and higher molecular weight that have a higher melting temperature. This behavior is confirmed by the report of Yang et al. [16] who produced extruded products of starch–emulsifier mixtures and observed that the transition temperatures increase after the extrusion apparently due to possible interactions between the components of the matrix.

Several authors point out that, during the extrusion process, the crystalline structure of the starch particles present in the food is weakened in addition to the splitting of the double helices of amylopectin [16,25]. These conformational changes uncover several active sites that favor the interactions between the protein and starch polymers generating three-dimensional structures with greater capacity to retain water [25].

The content of squid in the protein–carbohydrate matrix showed a significant effect (*p* < 0.05) on the enthalpy values (ΔH) (Figure 2). It was observed that increasing the squid content from 40 to 100% reduced the ΔH values in both non-extruded and extruded samples apparently as a result of a reduction in the protein–starch interactions due to the lower amount of available starch.

The data of Figure 2 indicate that the extrusion process significantly reduces (*p* < 0.05) the ΔH values; this can be attributed to exposure of the matrix of the components to high temperatures, cutting effort and high pressures during the process. Several authors point out that these conditions can modify the crystalline and amorphous regions and helical structure of the starch and the folding of the β-sheet structures of fibrillar proteins [26], thus reducing the amount of energy needed for phase transition [21,27]. Hu et al. [26] mention that the reduction in gelatinization enthalpies indicates loss of internal crystallinity and the destruction of amorphous regions and modifications of the internal double helix structure of amylose present in starch [28].

Although it has been reported that during the extrusion process the protein–carbohydrate interaction is favored, this type of complexes formed can be very varied and is associated with the type of proteins and carbohydrates present, as well as their ability to form bonds. The extrusion process can contribute to the formation of new peptide bonds between its amino groups and carboxyl groups, which are responsible for crosslinking and electrostatic, and hydrophobic interactions can also occur [29].

### 2.4. Molecular Identity

The data of Figure 3 indicate that the squid content in the samples increased the area under the curve of the 1615 and 1540 cm^−1^ bands that correspond to the functional groups amine I and amine II present in the squid proteins. These bands were not observed in the samples containing 100% of potato and corn due to their lower protein content, but a band of 1015 cm^−1^ was present and was associated with vibration of *O*-glycosidic bonds characteristic for foods rich in carbohydrates [23].

In the samples with 40, 60 and 80% of squid (Figure 3a–c), the extrusion process promotes changes in the functional groups reflected in a decrease in the intensity of 1615 and 1540 cm^−1^ bands which are responsible for the intra- and inter-molecular interaction of the protein structures, allowing the structure to unfold by reducing the protein–protein interactions and enabling a possibly of interaction with the starch present in the matrix. Likewise, the extrusion process caused an increase in the area at 1015 cm^−1^ in the samples that can be attributed to high pressure and temperature that promote the formation of *O*-glycosidic bonds; these bonds can form three-dimensional networks capable of trapping the water vapor generated during the extrusion process. This behavior is similar to an increase in the expansion values observed in extruded squid–starch samples [4].

These results are in agreement with a report of Widjanarko et al. [5] who used FTIR to analyze the mixtures of isolates of soy, wheat, corn and galactomannan protein isolates from konjac (*Amorphophallus konjac*) for production of a food bar [5]; the authors reported bands with displacements of 1630 to 1650 cm^−1^ corresponding to the amine group I and similar values were observed in our study in the presence of proteins. Similar displacements coinciding with the amine group I have been observed in other protein isolates [30,31]. The results of FTIR and DSC indicate that the squid content and the extrusion process significantly influence the protein–starch interactions; these interactions can be responsible for the results reported by Valenzuela-Lagarda et al. [4] who evaluated the effect of the squid content on the physical and morphometric properties of expanded snacks by extrusion noting that the products with better technological characteristics, expansion index (2.0–1.9) and bulk density (0.11–0.13) are obtained by using mixtures with 40 and 60% squid.

### 2.5. Deconvolution of FTIR Spectra

After deconvolution of the areas corresponding to the starch and amide I regions, the peaks appear at the wavelengths of 1047, 1022 and 995 cm^−1^ and 1520, 1540, 1624, 1650 and 1680 cm^−1^ (for the starch and amide I region, respectively). These coincide with that indicated by various authors, who have used this region of the IR spectrum in starch samples using three main modes with maximum absorbances at 1047, 1022 and 995 cm^−1^ [32,33]. The bands at 1047 and 1022 cm^−1^ were associated with the ordered and amorphous structures of starch, respectively [14]. The absorbance ratio 1047/1022 cm^−1^ has been used to quantify the degree of order in starch samples [33,34].

In Table 2, it can be seen that the increase in squid concentration from 40 to 60% in the samples increases the 995/1022 ratio, that is, the degree of starch structural disorder is favored; in the same way, in the samples of 40 and 60% of squid the extrusion process increases the degree of structural disorder. When comparing the values of 995/1022 for the samples with 100% potato and 100% corn, it can be observed that the corn starch has more ordered structures; in the same way the extrusion favors more disordered structures in the potato samples, while corn samples acquire a more orderly conformation.

Similar results were observed by Sevenou et al. [35], who observed a clear segregation existed between potato and amylomaize starches, defined by the highest values for the ratio 1047/1022 cm^−1^ and the lowest for the ratio 1022/995 cm^−1^, and wheat, maize, and waxy maize starches. The band at 1022 cm−1 is less pronounced in the potato and amylomaize than in wheat, maize and waxy maize. Similarly, Warren et al. [30], observed that when the native (and therefore ordered) starch in excess of water is subjected to a hydrothermal treatment in an amorphous form, there is a clear change of the maximum position from 1000 to 1022 cm^−1^, indicating that the Maximum position change from 1000 to 1022 cm^−1^ is the most dramatic change in the spectra as a result of changes in the ordered structure. Table 3 shows the peaks in the wavelengths after the deconvolution, showing peaks of 1520, 1540, 1624, 1650 and 1680 cm^−1^, corresponding to the amide I region. It can be observed that increasing the squid content increases the amplitude of the peak, which is attributed to the increase in the protein content in the sample. It is observed that the greatest amplitude is shown by the 1650 wavelength peaks, followed by 1624 cm^−1^.

These values coincide with the results presented by Guerrero et al. [31], who report peaks with wavelengths of 1654 and 1545 cm^−1^ and 1692, 1630 and 1525 cm^−1^ for the contributions of the helix α and β sheet, respectively; in the same way the author points out that the areas of the bands at 1624 cm^−1^, 1650 cm^−1^ and 1680 cm^−1^ increase, are maintained or decreased depending on the type of polysaccharide used; therefore, differences in the spectra of the mixtures indicate that the use of different polysaccharides resulted in the production of different molecular conformations. In all the samples studied, the broadband produced at 1650 cm^−1^ reflects the presence of regular residual structures; in this position, the α helices and the disordered structures are more likely to predominate in the mixtures. The fact that the contributions of the β sheet also provide important information on the content of the α helix simply reflects the inverse correlation between both contents of the structure in the protein database [36].

## 3. Materials and Methods

### 3.1. Materials

The frozen giant squid mantle (*Dosidicus gigas*) and potato flakes (*Solanum tuberosum*) were purchased from a supermarket in the city of Culiacán, Sinaloa and were kept at −18 °C until use. The corn (*Zea mays*) grain used was Pioneer variety 3015W obtained from a grain collection center in the city of Culiacán, Sinaloa.

### 3.2. Preparation of the Sample

The squid mantle was thawed and washed, and the visceral layer was removed, as well as the outer layer and outer and inner tunics. Subsequently, the squid mantle was cut into 1 cm cubes and dehydrated by convection at 65 °C/18 h in a ZECD NSF-2 stove model (Excalibur Dehydrators, Sacramento, CA, USA). The dehydrated squid, corn grains and potato flakes were crushed in a hammer mill model MD01 (JERSA, Mexico City, Mexico) and then crushed more finely using a mill model LM3100 (Perten Instrument, Stockholm, Sweden) to 40 mesh. The formulations of dehydrated squid (DS), potato flour (P) and corn flour (C) were prepared.

The ratio of potato/corn flours (RPC) remained constant at 5: 1 (*w*/*w*) and the proportion of squid was varied at 40, 60, 80 and 100%; 01 and 02 correspond to 100% of potato and 100% of corn, respectively. Each of the formulations were conditioned with distilled water to 15% humidity before the process. The wet mixtures were packed in polyethylene bags and stored (4 °C/ 8 h). The samples were tempered (25 °C/1 h) before extrusion.

### 3.3. Preparation of the SEE

For the development of SEE, a single screw laboratory extruder Model 20-CW-DN (Brabender Instruments, Inc., South Hackensack, NJ, USA) was used; a 19 mm diameter screw and the length to diameter ratio 20:1 were used; the nominal compression ratio was 3:1 and output the diameter was 3 mm [37]. The barrel of the extruder is composed of three heating zones adjusted to constant temperatures of 95, 110, and 130 °C. The screw speed was 200 rpm. The sample was fed to a mass flow of 30–40 g/min. The extruded products were equilibrated at ambient conditions (25 °C, RH = 65%) and stored in polyethylene bags until use [4].

### 3.4. Proximal Analysis and Mineral Content

The determinations of moisture, ash, protein (N × 6.25), dietary fiber (Megazyme International, Wicklow, Ireland) and lipids in the samples were performed according to the methodologies of the AOAC [38]. The carbohydrate content was calculated as a difference. The mineral content (K+, Na+, Ca++, Mg++, Zn ++, Cu ++, Fe ++ and Mn++) was determined by atomic absorption by a 200 40FS spectrophotometer model (Agilent, California, USA) and phosphorus (P^3+^) was assayed by a UV-visible spectrophotometer model 6705 UV/VIS (Jenway, Staffordshire, UK) [38]. Proximal analysis and mineral content analysis were performed on each of the samples in triplicate.

### 3.5. In Vitro Protein Digestibility (IVPD)

One hundred milligrams of a dried, defatted sample were weighed in triplicate. The weighted sample was incubated with 20 mL of HCl 0.1 N and 1.5 mg of pepsin at 37 °C/3 h. The mix was neutralized with 10 mL of NaOH 0.2 N; 40 mg of pancreatin was added in 7.5 mL of phosphate buffer (pH = 8) and samples were incubated at 37 °C/24 h; then, 700 mL of trichloroacetic acid 80% was added and the sample centrifuged at 3215 g for 10 min. The sediment was supplemented with 30 mL of distilled H_2_O, filtered and dehydrated at 45 °C/24 h. The dehydrated residue and the original sample were used to determine the protein content (micro Kjeldahl). IVPD was calculated according to Equation (1) [22].
(1)$$IVPD=\frac{Sample\;Protein-Residue\;Protein}{Sample\;Protein}\times 100$$

### 3.6. Calorimetric Properties

Thermal properties of various mixtures were determined in triplicate before and after the extrusion process. The methodology cited by Zeng et al. [39] was followed using a differential scanning calorimeter (DSC) model 2920 (TA Instruments, New Castle, DE, USA) equipped with a thermal analysis data station. A dry sample (2.5 ± 0.1 mg) of various types of flour was loaded in an aluminum tray with 4 μL of distilled water and sealed hermetically. The DSC was calibrated using indium (In^+3^) after a reference point was set up with an empty aluminum container. The trays with the samples were heated at a speed of 10 °C/min from 20 to 250 °C. The thermogram was constructed ranging from the initial temperature to 250 °C and melting temperature (Tm) and the enthalpy of fusion (ΔH) were calculated per dry weight of flour and expressed in J/g.

### 3.7. Molecular Identity

The molecular identity study was carried out in the mixtures in triplicate before and after the extrusion process following the methodology proposed by Widjanarko et al. [5]. A Fourier transform infrared spectrophotometer model FTIR 8400 S (Shimadzu, Kyoto, Japan) was used. The samples (0.01 g) were homogenized with 0.01 g of anhydrous KBr in a mortar. The mixtures were pressed by hydraulic vacuum (Graseby Specac, Orpington, UK) at 1.2 psi to obtain a transparent pellet. The samples were analyzed by transmission in a range of 4000 to 600 cm^−1^, collecting 50 sweeps at 4 cm^−1^ resolution. The results of the analysis included chemical structure, form of molecular union and functional groups corresponding to the chemical structure of the sample.

### 3.8. Deconvolution of FTIR Spectra

An FTIR spectrum was obtained for each of the samples, the subtraction criterion was a flat baseline in the 2500/2000 cm region. The spectra were corrected at the beginning in the starch and amide I regions (800–1200 and 1500–1700 cm^−1^, respectively) before applying the deconvolution using Origin 8.0 The assumed line shape was Lorentzian with an average width of 19 cm^−1^ and a 1.9 resolution improvement factor. The IR absorbance values at 1047, 1022 and 995 cm^−1^ and 1520, 1540, 1624, 1650 and 1680 cm^−1^ (for the starch and amide I region, respectively) were extracted from the spectra after the baseline correction and Deconvolution. To compare the IR spectra, a vector normalization function provided by the software (Origin 8.0) in the 800–1200 cm and 1500–1700 region was used. [36]

### 3.9. Statistical Analysis

The study was carried out under a completely random design with two factors; F1 was inclusion level of squid (F1 = 01, 02, 40, 60, 80, 100% squid, 01 and 02 correspond to 100% potato flour and 100% corn flour, respectively) and F2 was sample processing (sample without extruding and extruded); three repetitions were used per treatment. The analyzed response variables included in vitro protein digestibility, fusion temperature, enthalpy of fusion and changes in the molecular structure, molecular bonds and functional groups. When the analysis of variance was significant (*p* < 0.05), a Turkey comparison was performed (α = 0.05) [40]. The statistical package MINITAB version 17 was used.

## 4. Conclusions

Expanded snacks made with squid–potato–corn have high protein content (40–85%) and high biological value (>93%). The results of differential scanning calorimetry and Fourier transform infrared spectroscopy indicated that the extrusion caused conformational changes in the structures of proteins and starch in the mixtures of squid–potato–corn flours, promoting protein–starch interactions that may be responsible for physicochemical and technological properties of the snacks expanded by extrusion. This study demonstrates that it is possible to prepare a squid-based snack with greater nutritional value (protein) and physicochemical characteristics similar to commercial snacks.

## Figures and Tables

**Figure 1 molecules-26-02103-f001:**
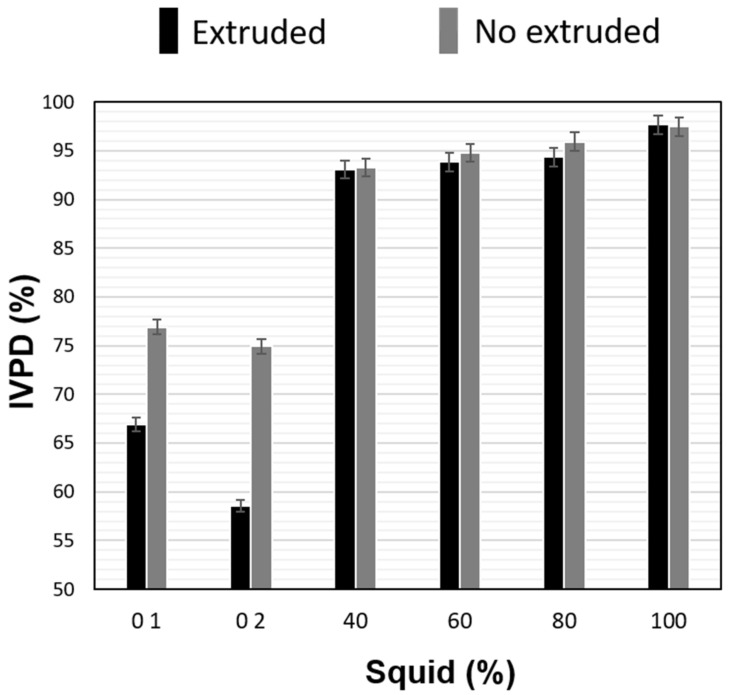
Effect of extrusion process and squid content on in vitro protein digestibility (IVPD) of expanded snacks by extrusion. The bars indicate the standard deviation of 3 samples.

**Figure 2 molecules-26-02103-f002:**
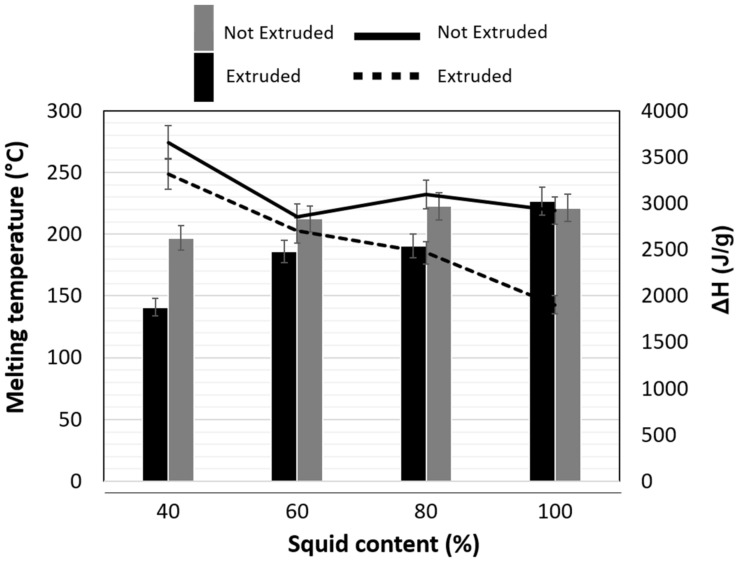
Effect of the squid content and the extrusion process on the calorimetric properties of the squid–potato–corn mixtures. Bar graph for melting temperature and line graph indicates enthalpy. The bars indicate the standard deviation of 3 samples.

**Figure 3 molecules-26-02103-f003:**
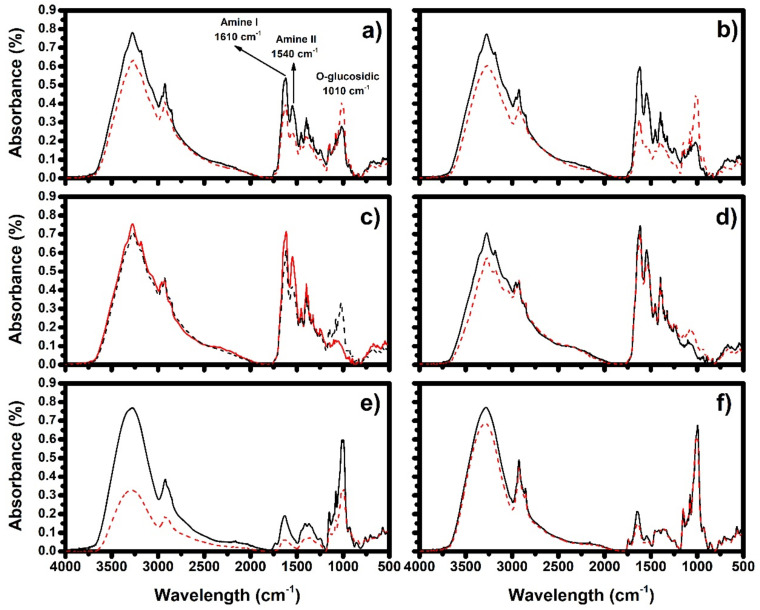
FTIR spectra of various mixtures of squid/potato/corn, not extruded (black line) and extruded (red dotted line). (**a**) 40% squid, (**b**) 60% squid, (**c**) 80% squid, (**d**) 100% squid, (**e**) 100% potato, (**f**) 100% corn.

**Table 1 molecules-26-02103-t001:** Proximal composition and minerals (% dry base) of snacks expanded by extrusion with different levels of squid inclusion.

Compound	Squid (%)					
	0_1_ *	0_2_ *	40	60	80	100
Moisture (%)	5.9 ± 0.2 ^b**^	6.52 ± 0.3 ^a^	4.32 ± 0.36 ^c^	4.82 ± 0.26 ^c^	5.68 ± 0.2 ^b^	5.27 ± 0.7 ^b,c^
Protein (%)	17.2 ± 0.7 ^e^	16.5 ± 0.4 ^e^	40.6 ± 1.52 ^d^	62.8 ± 1.7 ^c^	69.9 ± 1.1 ^b^	85.8 ± 1.5 ^a^
Lipids (%)	1.34 ± 0.3 ^b^	1.19 ± 0.4 ^b^	1.74 ± 0.4 ^b^	2.82 ± 0.6 ^a^	2.54 ± 0.5 ^a,b^	2.68 ± 0.2 ^a^
Carbohydrates (%)	65.8 ± 2.3 ^a^	66.2 ± 2.8 ^a^	42.3 ± 2.93 ^b^	20.16 ± 2.1 ^c^	10.9 ± 0.9 ^d^	5.64 ± 0.9 ^e^
Dietary fiber (%)	8.29 ± 2.0 ^a^	8.78 ± 1.8 ^a^	7.29 ± 1.24 ^a^	5.29 ± 1.8 ^a^	6.72 ± 0.6 ^a^	0.00 ^b^
Ashes (%)	2.89 ± 0.3 ^c^	1.21 ± 0.35 ^d^	3.63 ± 0.16 ^b^	4.09 ± 0.3 ^a,b^	4.23 ± 0.1 ^a^	3.73 ± 0.2 ^b^
Fe (mg/100g)	1.78 ± 0.4 ^a,b^	2.56 ± 0.78 ^a^	1.66 ± 0.12 ^a,b^	1.59 ± 0.2 ^b^	1.73 ± 0.2 ^a,b^	1.79 ± 0.3 ^a,b^
Mn (mg/100g)	0.46 ± 0.02 ^a^	0.26 ± 0.02 ^b^	0.22 ± 0.01 ^c^	0.20 ± 0.02 ^c^	0.27 ± 0.02 ^b^	0.29 ± 0.03 ^b^
Zn (mg/100g)	0.81 ± 0.01 ^e^	1.38 ± 0.01 ^d^	4.15 ± 0.05 ^c^	4.36 ± 0.1 ^b^	5.11 ± 0.3 ^a^	4.11 ± 0.04 ^c^
Cu (mg/100g)	0.29 ± 0.01 ^b,c^	0.28 ± 0.01 ^c^	0.34 ± 0.04 ^b^	0.39 ± 0.05 ^a,b^	0.42 ± 0.04 ^a^	0.45 ± 0.05 ^a^
Na (mg/100g)	92.2 ± 2.5 ^c^	92.45 ± 1.26 ^c^	268.7 ± 10.6 ^a,b^	265.9 ± 11.5 ^a,b^	250.9 ± 3.7 ^b^	273.7 ± 10.6 ^a^
K (mg/100g)	1087.5 ± 23.1 ^a^	1098.6 ± 17.4 ^a^	1006.1 ± 46.7 ^b^	1082.5 ± 56.3 ^a,b^	1078.0 ± 24.3 ^a^	1038.3 ± 32.9 ^a,b^
Ca (mg/100g)	53.6 ± 10.4 ^a,b^	53.8 ± 6.6 ^a^	46.5 ± 1.3 ^a^	46.5 ± 2.3 ^a^	52.19 ± 0.7 ^a^	42.87 ± 1.5 ^b^
Mg (mg/100g)	98.7 ± 2.8 ^d^	98.8 ± 1.5 ^d^	111.7 ± 0.7 ^c^	150.9 ± 1.4 ^a^	107.38 ± 2.7 ^c^	115.39 ± 0.6 ^b^
P (mg/100g)	758.2 ± 12.9 ^a^	662.6 ± 7.1 ^b^	608.3 ± 4.8 ^c^	623.8 ± 4.7 ^c^	628.7 ± 26.4 ^b,c^	616.28 ± 5.3 ^c^

Note: * 0_1_ and 0_2_ % correspond to 100% potato flour and 100% maize flour, respectively. ** Different letters in superscripts for each variable mean significant differences (Tukey, *p* ≤ 0.05).

**Table 2 molecules-26-02103-t002:** Absorbances after the deconvolution of the starch region of the different squid–potato–corn mixtures.

Sample	995 cm^−1^	1022 cm^−1^	1047 cm^−1^	Rate 995/1022 cm^−1^	Rate 1047/1022 cm^−1^
40% Squid NE	0.144	0.103	0.161	1.402	1.566
60% Squid NE	0.091	0.061	0.135	1.502	2.235
80% Squid NE	--	--	--	--	--
100% Squid NE	--	--	--	--	--
100% Potato NE	0.367	0.377	0.167	0.975	0.443
100% Corn NE	0.503	0.345	0.212	1.457	0.614
40% Squid E	0.218	0.270	0.126	0.808	0.466
60% Squid E	0.258	0.283	0.134	0.911	0.473
80% Squid E	0.0643	0.289	0.094	0.223	0.326
100% Squid E	--	--	--	--	--
100% Potato E	0.205	0.177	0.065	1.155	0.368
100% Corn E	0.417	0.395	0.177	1.054	0.447

* Note: NE: not extruded, E: extruded.

**Table 3 molecules-26-02103-t003:** Absorbances after the deconvolution of the amide I region of the different squid–potato–corn mixtures.

Sample	1624 cm^−1^	1650 cm^−1^	1680 cm^−1^	1520 cm^−1^	1540 cm^−1^
40% Squit NE	0.24799	0.29648	0.09178	0.10836	0.10172
60% Squit NE	0.26763	0.33159	0.09936	0.13769	0.12993
80% Squit NE	0.33556	0.37597	0.11486	0.18532	0.16496
100% Squit NE	0.32455	0.37964	0.11224	0.18954	0.15614
100% Potato NE	--	--	--	--	--
100% Corn NE	0.02584	0.15281	0.00439	0.01304	0.03574
40% Squit E	0.21708	0.18144	0.07765	0.0581	0.04407
60% Squit E	0.17454	0.10176	0.05827	0.00798	0.02526
80% Squit E	0.32227	0.26504	0.1026	0.11926	0.07978
100% Squit E	0.3154	0.32603	0.11334	0.1819	0.12079
100% Potato E	--	--	--	--	--
100% Corn E	--	--	--	0.01105	0.01591

## Data Availability

Data is contained within the article.

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
