# Peer review of "Interaction of Squid (Dosidicus giga) Mantle Protein with a Mixtures of Potato and Corn Starch in an Extruded Snack, as Characterized by FTIR and DSC"

_molecules, 2021, doi:10.3390/molecules26072103_

Round 1

Reviewer 1 Report

The abstract should be corrected. The last sentence is inadequate. Moreover too much overall information is included in the beginning and more result should be shoved.

Moreover, in the abstract author have stated that it is important to know the type of protein-carbohydrate interactions and their effect of sensory properties of snack. Why authors did not perform such study?

Line 77 Please correct the unit (-1) should be in upper appendix. Please also check in the whole manuscript for example line 94.

Authors did not include information about the number of repetitions of individual experiments.

It is a pity that authors did not determined others properties of snacks and not tired to explain or correlate FTIR or DSC results with different properties of snack.

It is not adequate to include the line number in Table 2.

Why in Table 3 authors did not include statistical evaluation of data.

Where is the title of Table 3?

Why under the Table 3 is the title of Table 2.

Reviewer 2 Report

The paper "Study of the interaction of the squid mantle protein (Dosidicus 2 gigas) and vegetable flour starch in an expanded snack by ex-3 trusion by FTIR and DSC" by José Luis Valenzuela-Lagarda et al. reports on the study of the interaction of the squid  mantle protein with the starch of potato and corn flours in  "snack expanded by extrusion" foods.

The subject is  interesting, the science is solid and  the work well performed. The experimental part is well described and convincing. However,  this referee would have expected more information on "interaction", as suggested by the title, and more accuracy in the preparation of the manuscript. 

At one side, interactions should be detailed in an explicit way: are chemical interactions ? physical ? what kind of complexes forms ? what kind of compound of high molecular weight ? Several references are indicated, but this is the central point of the paper and should be deeply investigated. On the other side, all the experimental data should be reported (e.g., calorimetric data should be shown), to help the reader to assess the conclusions.

Other points:

- Table I: in a few cases, errors appear to be inconsistent with the reported results, or, maybe, the number of significant digits is inconsistent with the experimental accuracy.

- Fig 2: In the figure (and in the text), histogram and line representations are not assigned to any data: who is with whom ?

- Fig. 3: For each frame, both lines are indicated as gray in the caption.. so what ? what is the difference between the two curves ?. Moreover, what means "extrusion (gray line)  and after extrusion (gray line)"

- Table 2 and 3: are the reported digit significant ?  are you sure the error is that low (around 0.005 %) ?

- Table 3: the  description is missing. And the text around Table 3 seems to be corrupted.

- at line 245: what are "protein double helices" ??

Reviewer 3 Report

The title should be changed to:

“Interaction of squid (Dosidicus giga) mantle protein with a blend of potato and corn starch in an extruded snack, as characterized by FTIR and DSC”.

It is important the leave the term expanded as the authors are not showing physical data of expansion index or porosity. The authors mention expansion index (EI) and high values of bulk density (BD) in the introduction as relevant physical properties for expanded extruded snacks and certainly these physical measurements should be done in this study. I recommend, if they still have samples for all the formulations studied, to do EI, BD and if they have a gas pycnometer also do true density and apparent porosity as affected by different concentrations of squid mantle in the extruded snack formulations.

L21-22 erase the statement “This could be complemented with something from the conclusions at the end of the manuscript” and describe in detail what are those conclusions.

L27 erase >20% of recommended daily intake. It is confusing at least, unless you elaborate what are you trying to convey.

L31 change “important for” to: abundant in

L290 Repeated words in Figure 3 title. Please clarify which lines correspond to extruded and non-extruded samples.

On Table 2 and 3 correct all the terms “squit” for” squid

Table 2. Rephrase the title of this table.

Table 3. Why not adding this data in table 2 and delete Table 3. Also, if you need to keep this table add an appropriate title and move the conclusions indicated now in the current Table 3 title to the corresponding section of the manuscript.

L356 Erase the Title used in Table 2 appearing at the bottom of Table 3.

Author Response

Dear reviewer, thank you very much for the detailed review of this manuscript, in the same way, the very correct observations are appreciated, which will allow us to improve the quality of this document. Each of the comments has been analyzed, and the suggested changes have been made, in the same way the observations made and the way in which each one of them was addressed are listed below, again thank you very much.

Round 2

Reviewer 1 Report

Authors corrected manuscript accordingly. 

Reviewer 2 Report

The paper has been modified as suggested. The paper can be published as it is.